# Hospital-at-home care in Singapore: A review of overseas protocols and guidelines to support implementation and policy redesign (systematic review)

Clive Goh[1][⊕][*], Jun Jie Soh[1][⊕], Valerie Ng[2][‡], Yi Feng Lai[1,2][‡]

1 Department of Pharmacy & Pharmaceutical Sciences, National University of Singapore, Singapore,
2 MOH Office for Healthcare Transformation, Singapore, Singapore

⊕ These authors contributed equally to this work.
‡ VN and LYF also contributed equally to this work.
* cgohjiaj@gmail.com

## Abstract

### Objective

Hospital-at-Home (HaH) is a care model providing acute, hospital-level care to patients in their own homes. It is gaining traction worldwide and could become an integral part of standard healthcare in the future. However, for countries like Singapore, implementation inefficiencies prevent the optimal uptake and establishment of HaH. Currently, there are no standardised guidelines guiding healthcare providers on effective implementation. Thus, our objective is to distil information from existing guidelines worldwide to collate the best practices for HaH implementation.

### Methods

The systematic review is according to the Preferred Reporting Items for Systematic Review and Meta-Analysis (PRISMA) 2020 checklist. A literature review across three databases and an Advanced Internet Search was performed to collect guidelines that included recommendations on HaH implementation requirements. Two authors independently extracted recommendations. Two reviewers independently assessed guideline quality using the Appraisal of Guidelines for Research and Evaluation II Instrument (AGREE II), which consisted of twenty-three items across six domains. Overall guideline quality was calculated as the total points from all six domains as a percentage over maximum points, and guidelines with overall scores of 50% or more were deemed high quality.

**Data availability statement:** All relevant data are within the manuscript and its Supporting Information files.

**Funding:** The author(s) received no specific funding for this work.

**Competing interests:** The authors have declared that no competing interests exist.

## Results

Fourteen guidelines and seven sections were identified, which covered the following topics: inclusion & exclusion criteria, admission process, clinical handover, discharge, team structure, partnerships with external stakeholders, and medication administration. Key observations underline deficiencies in addressing staff safety, appropriate medical supply storage, and admission after hours. The main discrepant recommendations included self-discharge, team structure, partnerships with primary care providers, and medication self-administration. Methodological quality of guidelines varied, with overall AGREE II scores ranging from 38.5% to 58.4%. Individual domain scores were consistently low for Rigour of Development and Editorial Independence. Despite low scores in these domains, all fourteen guidelines were deemed appropriate for information extraction.

## Conclusion

Despite inconsistencies among guidelines, a set of streamlined recommendations were consolidated. In Singapore, addressing home environment constraints, fostering stronger partnerships with community providers, and leveraging on multidisciplinary care can enhance the feasibility and sustainability of these HaH recommendations. Policy redesign should focus on further stratifying patients based on home suitability, leveraging on technology to support clinical handovers or collaborations, and investing in multidisciplinary training to strengthen workforce capabilities. In all, healthcare providers around the world should consider contextualising these recommendations within local socioeconomic and healthcare contexts for optimal HaH implementation.

## Introduction

Hospital-at-Home (HaH) is a care model delivering acute, hospital-level care to patients in their own homes, typically for a specified duration, as a substitute for acute inpatient admission [1,2]. With the Covid-19 pandemic exposing healthcare system vulnerabilities worldwide, many countries, including Singapore, are turning to HaH as a viable alternative to inpatient care [3,4]. HaH has been shown to improve resource allocation, patient satisfaction, and patient-centred care [5–7].

HaH has a relatively long history in countries such as Australia [8]. Moreover, countries like the UK [9], Spain [10], and Canada [11] have tailored their services to specific conditions or patient groups. These could potentially offer implementation insights.

However, heterogeneity of guidelines globally reveal varying degrees of structuring HaH services [12,13], posing challenges for optimal model identification. Despite increasing interest in HaH models, implementation remains difficult in countries lacking national standards governing this practice [14,15]. For countries like Singapore, where HaH development is in its infancy, HaH faces implementation hurdles due to

unfamiliarity and limited experience [16,17]. Even for countries with established programmes like Spain [18], inefficient implementation hinders widespread adoption, necessitating policy redesign.

Despite existing systematic reviews comparing global HaH efficacy and safety, no 'gold standard' approach exists to guide implementation [19]. A standardised guide is therefore essential to facilitate optimal HaH uptake and advancement. **Our goal is to appraise guideline quality, make cross-country comparisons, reconcile differences, and propose best practices for HaH implementation,** culminating in a comprehensive implementation guidebook.

## Methods

This systematic review was completed according to the Preferred Reporting Items for Systematic Reviews and Meta-Analyses (PRISMA) 2020 (see S3 Table).

### Data sources

This literature review utilised two search methods: searches in three databases (PubMed, Scopus, and Web of Science) conducted on 9 October 2023, and an Advanced Internet Search performed on 7 December 2023 to include guidelines not indexed in these databases. Both searches used predefined search terms and inclusion/exclusion criteria. They were limited to papers published in the last 10 years. There was no registered protocol for this study.

### Search strategy

Two independent reviewers selected literature focusing on guidelines, recommendations, and policies for successful HaH implementation. Eligible studies defined HaH as the provision of acute short-term care to patients at home instead of inpatient hospital care [20–22], specifically for adult patients in geriatric, internal medicine, and general medicine specialties. Internal, family, or general medicine were defined as managing conditions beyond single-organ specialties or subspecialties. This involved more generalised, multi-system disease conditions.

Excluded were papers focusing on chronic illnesses, outpatient, transitional, or self-care interventions, and those lacking concrete recommendations for HaH implementation. The focus was to establish a stable programme covering the general population, before considering specialty care.

Table 1 details the full inclusion/exclusion criteria, while the complete search syntax can be found in S1 Table.

### Data extraction

Fig 1 displays the PRISMA flow diagram for search results. Database findings were screened for duplicates via Covidence (a tool for systematic reviews). Article selection involved two stages: initial screening of titles and abstracts by two independent reviewers, followed by full-text screening based on inclusion/exclusion criteria. Discrepancies were resolved through consensus discussions.

**Table 1. Inclusion/exclusion criteria.**

| Inclusion criteria | Exclusion criteria |
|---|---|
| Provides recommendations of specific guidelines, standard-operating-procedures, and/or policies for implementation of HaH. | Interventions involving chronic illnesses or provision of long-term, outpatient, post-discharge care, or self-care. |
| Demonstrate HaH as the provision of acute or subacute care to patients at home as an alternative to inpatient hospital care. | Other areas of care such as paediatrics, psychiatry, palliative, etc. |
| Involves patients from the geriatric, internal medicine and/or general medicine specialties. | Literature not giving any concrete recommendations or guidance for implementation of HaH. |
| Published in English. | Published prior to 2014. |
| Published between January 1, 2014 through October 1, 2023. | |

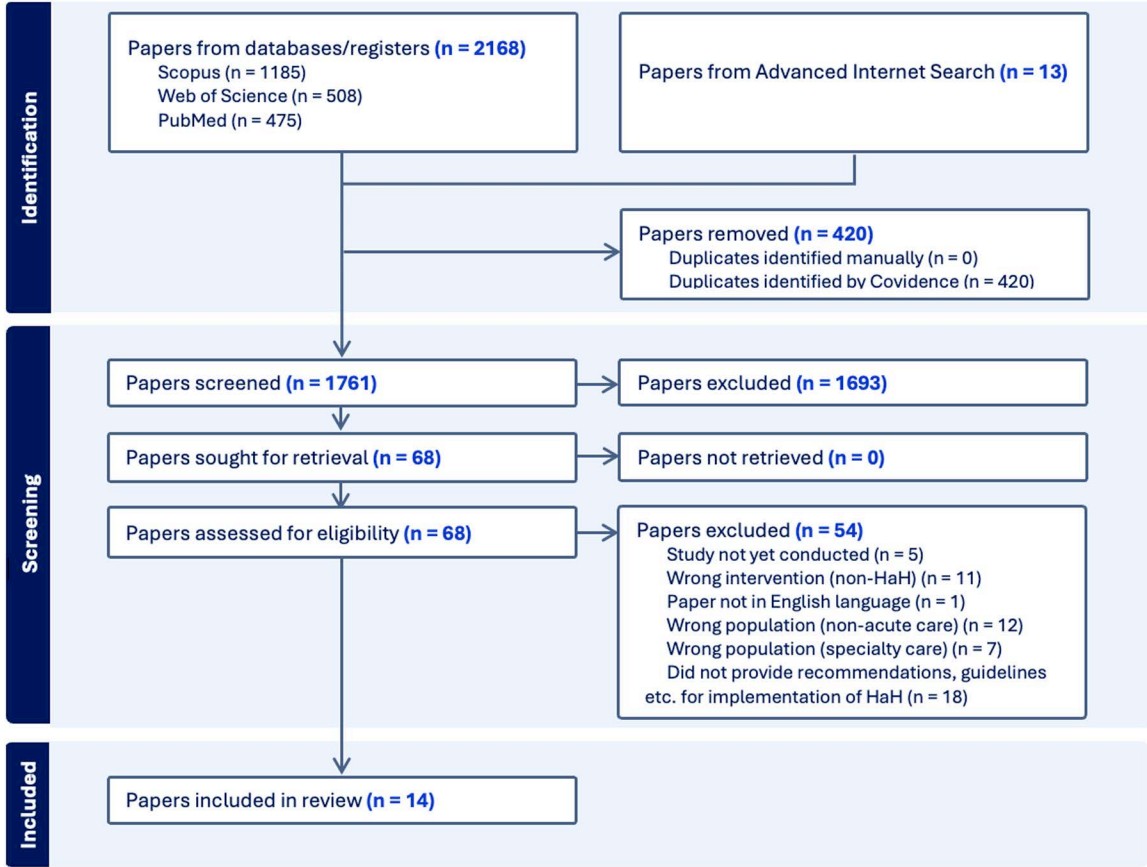

**Fig 1. PRISMA flow diagram.**

The Advanced Internet Search focused on five regions: with established HaH programmes (Australia, Canada, Spain, UK, USA). These countries were selected due to their well-developed HaH programmes, with the US, UK, and Australia each having over a decade of experience in HaH implementation. Countries with established programmes but predominantly non-English publications, such as France and Taiwan, were excluded due to language and translation barriers. The first 50–60 listings were taken to identify appropriate organisations and sites, then navigated to identify more relevant information. An additional 10–20 results were explored to ensure methodological relevance.

A pre-established template consisting of twelve components was employed to extract any and all recommendations provided from the selected guidelines (see S2 Appendix). Recommendations were summarised to highlight common and notable practices for effective HaH implementation.

## Quality assessment

Methodological quality was evaluated using the Appraisal of Guidelines for Research and Evaluation II (AGREE II) instrument, consisting of twenty-three items across six domains [23,24]. Items were rated on a seven-point Likert-type scale, using criteria in the User's Manual. Domain scores were calculated as a percentage of the maximum possible score. Two independent reviewers assigned scores to each item, resolving discrepancies in scores exceeding two points through

consensus discussions. Guidelines were considered high quality if the overall score (total points from all six domains as a percentage over maximum points) was 50% or higher [25,26].

## Results

The database search yielded 1,748 citations. After removing duplicates and applying inclusion/exclusion criteria, only one guideline was selected (see S4 Table). For the Advanced Internet Search, thirteen guidelines were identified, resulting in a total of fourteen guidelines for analysis.

### Guideline characteristics

Table 2 summarises key characteristics of the fourteen guidelines. The guidelines originate from the UK (five) [27–31], Australia (four) [32–35], USA (two) [36,37], Spain (two) [38,39], and Canada (one) [40]. All were published within the last ten years, except for "Hospital in the Home Guidelines" by Victoria State Government (2011) [34]. This paper was retained as the guidelines are still in use and cited on the state government's Department of Health website, last updated in 2022 [41].

**Table 2. Characteristics of Guidelines.**

| Article Name | Author | Year of Publication | Country |
|---|---|---|---|
| Adult and Paediatric Hospital in the Home Guideline [32] | NSW Health | 2018 | Australia |
| Policy Guideline<br>Hospital in the Home Policy Guideline [33] | SA Health | 2017 | Australia |
| Hospital in the Home Guidelines [34] | Victoria State Gov | 2011 | Australia |
| Hospital in the Home [35] | Queensland Gov | 2023 | Australia |
| Making the most of virtual wards, including Hospital at Home [27] | NHS | 2023 | United Kingdom |
| Guidance on Pharmacy Services and Medicines Use within Virtual Wards (including Hospital at Home) [28] | NHS | 2022 | United Kingdom |
| Bringing hospital care home:<br>Virtual Wards and Hospital at Home for older people [29] | BGS | 2022 | United Kingdom |
| Supporting information<br>Virtual ward including Hospital at Home [30] | NHS | 2022 | United Kingdom |
| Hospital at Home<br>Guiding Principles for Service Development [31] | Healthcare Improvement Scotland | 2020 | United Kingdom |
| Home Hospital Pharmacy Playbook<br>A Comprehensive Guide to Designing your Home Hospital Pharmacy Services [36] | ASHP, Mayo Clinic | 2022 | United States of America |
| Acute Care Delivery at Home [37] | TRACIE | 2021 | United States of America |
| Organisational model for Hospital at Home in Catalonia<br>Alternative to conventional hospitalisation [38] | Catalan Health Service | 2020 | Spain |
| Developing new hospital-at-home models based on Comprehensive Geriatric Assessment: Implementation recommendations by the Working Group on Hospital-at-Home and Community Geriatrics of the Catalan Society of Geriatrics and Gerontology [39] | Sociedad Española de Geriatría y Gerontología | 2022 | Spain |
| Bringing Acute General Internal Medicine Outside of Hospital Walls [40] | Canadian Journal of General Internal Medicine | 2022 | Canada |

**Abbreviations:** ASHP: American Society of Health-System Pharmacists; BGS: British Geriatrics Society; NHS: National Health Service; NSW: New South Wales.

## Guideline appraisal

Table 3 presents AGREE II scores for the fourteen guidelines. Overall scores ranged from 38.5% to 58.4%, with seven guidelines scoring 50.0% or higher [27,29,31,33–35,39]. The top-scoring guideline was "Hospital at Home: Guiding Principles for Service Development" by Healthcare Improvement Scotland (58.4%) [31], followed by "Bringing Hospital Care Home: Virtual Wards and Hospital at Home for Older People" by BGS (56.2%) [29]. The highest-scoring domains were "Scope and Purpose" (90.9%) and "Clarity of Presentation" (66.1%), while "Editorial Independence" (8.6%) and "Rigour of Development" (16.6%) scored lowest.

**Table 3. AGREE II Scores (%) of Guidelines.**

| Guideline (Author & Year) | Domain 1: Scope and Purpose | Domain 2: Stakeholder Involvement | Domain 3: Rigour of Development | Domain 4: Clarity of Presentation | Domain 5: Applicability | Domain 6: Editorial Independence | Overall Quality of Guideline | Guideline Recommended for use | Quality (High/Low) |
|---|---|---|---|---|---|---|---|---|---|
| NSW Health, 2018 [32] | 100% | 38.9% | 9.4% | 77.8% | 54.2% | 4.2% | 49.7% | Recommended | Low |
| SA Health, 2017 [33] | 100% | 50% | 16.7% | 63.9% | 43.8% | 8.3% | **50.3%** | Recommended | **High** |
| Victoria State Gov, 2011 [34] | 100% | 66.7% | 11.5% | 72.2% | 41.7% | 8.3% | **51.2%** | Recommended | **High** |
| Queensland Gov, 2023 [35] | 100% | 41.7% | 18.8% | 63.9% | 50% | 0% | **50.3%** | Recommended, with modifications | **High** |
| NHS, 2023 [27] | 100% | 41.7% | 17.7% | 94.4% | 35.4% | 0% | **51.2%** | Recommended, with modifications | **High** |
| NHS, 2022a [28] | 100% | 63.9% | 15.6% | 50% | 35.4% | 8.3% | 48.8% | Recommended, with modifications | Low |
| BGS, 2022 [29] | 91.7% | 83.3% | 29.2% | 69.4% | 37.5% | 4.2% | **56.2%** | Recommended, with modifications | **High** |
| NHS, 2022b [30] | 75% | 13.9% | 16.7% | 66.7% | 41.7% | 4.2% | 43.2% | Recommended, with modifications | Low |
| Healthcare Improvement Scotland, 2020 [31] | 100% | 44.4% | 39.6% | 86.1% | 43.8% | 0% | **58.4%** | Recommended | **High** |
| ASHP, Mayo Clinic, 2022 [36] | 52.8% | 41.7% | 6.3% | 69.4% | 35.4% | 0% | 39.8% | Recommended, with modifications | Low |
| TRACIE, 2021 [37] | 72.2% | 55.6% | 7.3% | 13.9% | 41.7% | 0% | 38.5% | Recommended, with modifications | Low |
| Catalan Health Service, 2020 [38] | 100% | 22.2% | 13.5% | 72.2% | 50% | 0% | 47.5% | Recommended, with modifications | Low |
| Sociedad Española de Geriatría y Gerontología, 2022 [39] | 94.4% | 36.1% | 16.7% | 77.8% | 14.6% | 70.8% | **50%** | Recommended, with modifications | **High** |
| Canadian Journal of General Internal Medicine, 2022 [40] | 86.1% | 41.7% | 13.5% | 47.2% | 56.3% | 12.5% | 47.2% | Recommended, with modifications | Low |
| Overall Domain Score | 90.9% | 45.8% | 16.6% | 66.1% | 41.5% | 8.6% | | | |

## Guideline recommendations

The guidelines were categorised into twelve components and forty-six sections, with seven key sections selected for evaluation based on their operational and organisational significance in HaH programmes, providing insights distinct from traditional hospital settings. The remaining sections covering more basic parameters are detailed in S2 Appendix.

## Inclusion & exclusion criteria

Ten guidelines addressed inclusion criteria for HaH admission, while nine addressed exclusion criteria. Both criteria can be classified into medical, functional, and social categories.

Medical requirements: Eligibility includes acute or subacute conditions warranting hospital admission [31–33,35,37–40], and requiring minimally once daily care team visits [27,33,35]. Patients must be clinically stable and not require continuous assessment or treatment [32,33,38–40]. One guideline included patients opting against hospitalisation for personal, psychological, or social reasons [33]. Conversely, five guidelines excluded non-acute care patients [27,33–35,37], including those under post-discharge care [27,33,35]. Patients requiring care complexity exceeding HaH capacity [32,35], or intensive care for conditions like acute coronary syndrome, stroke, and suspected neck or femur fractures were excluded [31,40]. Mental health conditions [35], behavioural problems [38,40], and unclear provisional diagnoses were deemed unsuitable [32].

Functional requirements: Patients were required to be competent in self-managing their condition or have a caretaker to do so [32,33,38–40]. Consent for enrolment was emphasised by six guidelines [32,33,35,38–40]. Patients demonstrating poor compliance with care were unsuitable [32].

Social requirements: The care location must be safe and suitable for both patients 31–33,35,38,39 and HaH staff [33]. Guidelines mandated appropriate clinical supply storage (e.g., refrigerators) [35], and reliable communication devices (e.g., landline, phones) at home [32,33,35,38,39. Patients residing far from the hospital were excluded [32].

## Admission process

Eight guidelines offered admission process recommendations. Three specifically addressed determining patient eligibility for HaH prior to admission [32,35,37]. Upon admission, one guideline emphasised providing comprehensive operational information about HaH to patients [32]. Two guidelines suggested documenting the clinical decision to admit, including patient information and consent [32,35]. Two guidelines highlighted the importance of aligning HaH documentation tools with other healthcare providers to prevent duplication or omission [34,35].

Regarding clinical review, one guideline recommended conducting it within 24 hours of HaH admission [32]. Consensus among multiple guidelines emphasised developing a holistic personalised care plan, considering clinical/medication risks, home environment, and patient aggression [32–35,37,39]. Assessments can be conducted at the patient's home [27,34,37].

Regarding admission timing, one guideline specified response times from patient identification to HaH admission to be within 24 hours from emergency services and 48 hours from hospital wards [38]. Another guideline suggested the possibility of after-hours admission [32].

## Clinical handover

Eight guidelines detailed clinical handover procedures, emphasising patient movement into and out of HaH and stressing the importance of strong relationships with primary health networks and community services [29,31–35,38,39]. One guideline recommended formal documented handover policies and processes [32].

Regarding care transfer, three guidelines emphasised formal patient discharge from one facility before admission to another [32–34]. HaH services should plan appropriate referrals and engagement with other programmes, maintaining

case management until clinical handover [33–35]. Primary care providers should be notified of HaH admission within 24–48 hours [35].

### Discharge

Six guidelines outlined HaH discharge processes. Four recommended allocating an estimated discharge date, communicating it to patients, and reviewing it daily [32–34,39]. Discharge should not be delayed without justification [32,34]. Referrals to community–based services should be initiated as soon as clinically appropriate before discharge [32–34,39]. Two guidelines included options for self-discharge [32,33].

Upon discharge, five guidelines recommended sending a shared care plan [32], or discharge summary to the receiving service (e.g., general practice (GP), community care) [28,33–35], within 24 hours [32]. They also advised providing patients and carers with documents containing plain-language information on medication management, follow-up appointments, and community support contacts.

Post-discharge, one guideline suggested referral to pharmacists to enhance medication-related communication and reduce harm [28].

### Team structure

Seven guidelines outlined team structures within HaH programmes, proposing two main models. The first model involves HaH staff delivering care in both hospital and home settings [32,35]. The second model divides the team into a hospital-based group responsible for care plans and intensive care, and a home-visiting team comprising community paramedics (CPs) [40]. CPs would assist in patient assessment and management, relaying care plan suggestions directly to HaH clinicians.

Two subcategories were also proposed: standalone teams employed specifically for HaH care, and integrated teams comprising hospital, primary, and community care staff [32,35]. The integrated teams would manage patients from HaH alongside their core roles.

Four guidelines specified team compositions of medical, nursing, pharmacy, allied health (e.g., physiotherapists, occupational therapists), and support staff [32,37–39]. Two guidelines recommended integrating CPs [31,37], while another outlined the support roles required, including administrative staff and operational managers [37].

### Partnerships with external stakeholders

Eight guidelines offered recommendations on partnerships with other programmes and external service providers. Two guidelines suggested integrating HaH into strategic planning with primary health networks [30,32], while two advocated partnerships with primary service providers, specifically GPs [32,33]. Interestingly, one guideline discouraged GP partnerships, advocating for a clear distinction between HaH and primary care [35]. Other partnerships included community nursing services [32–34], chronic disease programmes [32,33], allied health services (including community pharmacy leaders) [28,33], intermediate care services [33], and admission avoidance programmes [27,29].

### Medication administration

Five articles discussed medication administration methods in HaH, with most advocating for self-administration. However, two guidelines recommended that all HaH medications be administered by staff with appropriate competencies and clinical privileges [34,35].

Among guidelines supporting self-administration, two recommended conducting risk assessments to determine patient suitability [28,32]. These guidelines specified that certain medications, such as parenteral medications, could be self-administered. For more complex regimens, additional training could be provided. This involved written

instructions, education on administration techniques or monitoring for adverse effects, and enhancing adherence via multi-compartment compliance aids [28,32].

Close support from the HaH clinical team was highlighted, including regular medical reviews via phone calls or face-to-face assessments, as well as 24-hour on-call medical staff support [32]. Additionally, one guideline suggested using virtual nurse validation via camera for medication administration [36].

For controlled medications, one guideline emphasised the need for appropriate waste disposal containers at patient homes, with administration and disposal processes documented and witnessed by nurses [36].

## Discussion

This review analysed fourteen guidelines on HaH implementation across various countries, representing the largest and most comprehensive systematic review to date on this topic. While most guidelines followed a standard framework of recommendations, with common elements such as inclusion/exclusion criteria and admission processes, recommendations were often inconsistent, hindering the establishment of a general consensus for implementation. This discussion aims to streamline recommendations, highlighting contrasting, surprising, or critical points for further evaluation.

### Recommendations

Table 4 describes recommendations synthesised from all fourteen guidelines.

### Inclusion & exclusion criteria

Addressing staff safety is a crucial but often overlooked aspect in most guidelines. In HaH, the dynamic shifts as healthcare staff become "guests" in patients' homes [42]. This change, along with safety concerns like hygiene and space limitations that are harder to control compared to a hospital environment, necessitate additional regulations to safeguard healthcare workers [15].

Reliable communication devices and adequate medical supply storage is pivotal for patient safety. Given HaH's emphasis on patient self-management, communication devices are essential to bridge communication between providers and patients. Similarly, proper medication storage is vital. As spatial limitations are a significant concern for many HaH patients [43], it is crucial to consider providing dedicated clean spaces, like a medical cabinet, to ensure proper medication management and storage.

Admission to HaH for personal, psychological, or social reasons was excluded as it diverges from HaH's primary objectives. With healthcare resource scarcity and HaH's reliance on hospital resources, efficient utilisation is crucial. Resources should prioritise patients in greatest need, redirecting non-acute indications to alternative options.

### Admission process

After-hours admission was included despite only being suggested by one guideline, due to its critical role in providing continuous care and maximising patient outcomes. While this may require additional resources, HaH programmes from Spain, UK, and Canada have already successfully adopted an after-hours, 24-hour system via integration with existing standby services [44]. Programme planners should assess operational constraints against patient needs to determine the feasibility of after-hours services.

### Clinical handover

The measures created for clinical handover promote service integration and ensure a seamless continuum of care for HaH patients. They align with initiatives in various countries aimed at streamlining care services, such as having dedicated GP providers for patients [45–48]. Building robust relations with stakeholders can alleviate operational constraints, ease transitions in and out of HaH, and simplify the identification of suitable candidates.

**Table 4. Overall recommendations.**

| Area of HaH implementation | Recommendation |
|---|---|
| Inclusion Criteria | Patients with acute/subacute conditions; |
| | Require a minimum of once daily care team visits; |
| | Clinically stable; |
| | Able to manage their own condition/have a suitable caretaker to do so; |
| | Obtained consent; |
| | Home environment that is safe for both patient and staff; |
| | Have access to reliable communication devices; |
| | Have access to appropriate storage for medical supplies at home. |
| Exclusion Criteria | Patients requiring non-acute care, including post-discharge care; |
| | Conditions requiring more intensive care; |
| | Care complexity exceeding HaH capacity; |
| | Mental health conditions and/or behavioural problems and/or suicide risk |
| | Unclear provisional diagnosis; |
| | Poor compliance with care; |
| | Living in a remote location from the hospital. |
| Admission Process | Determine patient eligibility prior to admission; |
| | Provide patient with comprehensive HaH information upon admission; |
| | Document patient information and consent, using tools aligned with other healthcare providers. Utilise existing documentation to minimise duplication or omission; |
| | Conduct a clinical review and develop a holistic, personalised care plan (can be done in a clinical setting or at the patient's home); |
| | Ensure timing of admission into HaH is within 24 hours from emergency services or 48 hours from hospital wards; |
| | Set up after-hours services. |
| Clinical Handover | Establish strong working relationships and clear lines of communication with other healthcare and community stakeholders; |
| | Ensure formal discharge from one facility before subsequent admission to another; |
| | Maintain HaH as the main coordinator of care with other programmes until handover. |
| | Develop formal, documented handover policies and processes; |
| Discharge | Determine and inform the patient of an estimated date of discharge, to be reviewed daily; |
| | Make referrals to community-based services as soon as clinically appropriate, prior to discharge; |
| | Send a shared care plan or discharge summary to the receiving service during discharge; |
| | Provide patients and carers with necessary information (in layman's terms) to facilitate care after discharge; |
| | Refer patients to pharmacists for medication counselling. |
| Team Structure | Team structure:<br> - Segregating home care and hospital teams **OR**<br> - Allow HaH teams operate in home and hospital care.<br>Team origin:<br> - Standalone team (i.e., dedicated HaH staff) **OR**<br> - Integrated team (i.e., combines staff from hospital, primary, and community care).<br>**The chosen team should depend on the programme's requirements.** |
| | Team composition:<br> - Medical;<br> - Nursing;<br> - Pharmacy;<br> - Allied health professionals (e.g., physiotherapists, occupational therapists, etc);<br> - Support staff (administrative, operational managers, etc.);<br> - Community paramedics (CPs). |

*(Continued)*

 

**Table 4.** (Continued)

| Area of HaH implementation | Recommendation |
|---|---|
| Partnerships with External Stakeholders | HaH should be included in strategic planning with primary health networks; |
| | HaH should partner with other healthcare providers (e.g., GPs, community nursing services, chronic disease programmes, allied health services, intermediate care services, and other admission avoidance programmes). However, the HaH service should be held accountable for the overall patient care safety and quality. |
| Medication Administration | Encourage medication self-administration (including parenterals), with the following protocols:<br> - Conduct risk assessment to determine patient suitability;<br> - Provide training on medication administration techniques, adherence, and monitoring of adverse effects;<br> - Offer close support with regular medical reviews via phone calls or face-to-face assessments. |
| | When self-administration is infeasible, establish clear processes for staff intervention;. |
| | For controlled medications, implement appropriate processes for documentation and disposal in patients' homes (e.g., provision of appropriate waste containers). |

## Discharge

Recommendations for discharge procedures mirror traditional inpatient care in enhancing patient flow, resource management, care continuity, and reducing transition-related errors.

A noteworthy consideration not recommended is self-discharge. In hospitals, discharge decisions are predominantly clinician-led to meet medical and safety standards. While allowing self-discharge promotes patient autonomy and shared decision-making, it is associated with higher readmission and mortality rates [49]. Self-discharge typically stems from perceived negative experiences, care dissatisfaction, and personal or financial constraints [50]. HaH teams should reassess care quality should self-discharge occur.

Referrals to pharmacists for medication counselling optimises medication outcomes. Pharmacist-led medication reconciliation programmes at hospital transitions reduce mortality, readmissions, and emergency visits. These offer significant benefits for long-term patient outcomes [51].

## Team structure

Various viable team structures were proposed for HaH. The chosen team structure should align with programme needs, considering factors like patient volume, acuity, staff numbers, and visit frequency [30]. Integrated teams may suit smaller HaH models, where service provision is shared between hospital and community care staff. Furthermore, during periods of high demand, they may rotate between HaH and their primary care roles as needed [32]. Regardless of structure, additional safeguards and policies are essential to ensure effective medical management, communication, and handover processes.

Nevertheless, it is recommended that all HaH teams include the core members described in Table 4. This composition mirrors traditional hospital teams, with added support personnel for overall operational efficiency. A novel recommendation involves integrating CPs to supplement nursing roles [52]. CPs can serve as 'physician extenders' in acute and home care, assisting in patient assessment, medication administration, physical exams, and managing clinical deterioration [53].

## Partnerships with external stakeholders

While guidelines varied in their recommendations for partnerships with external service providers, including one discouraging alignment with primary care providers like GPs, our approach differs. Although centralising care solely through HaH may streamline service provision, it overlooks potential challenges such as geographical barriers, peak demand periods, and the need to expand to different service types [34]. Therefore, partnerships with other healthcare providers should be emphasised to ensure continuity of care and optimise resource allocation across providers. Planners should ultimately

maintain HaH's primary accountability for the patient, with clear interdisciplinary agreements, transparent remuneration strategies, and shared decision support tools to enhance care delivery [32].

## Medication administration

Despite guidelines discrepancies, self-administration should be adopted for its benefits in enhancing patient knowledge, autonomy, and adherence [54], alongside reduced care costs and lower risks of healthcare-acquired infections [32]. Staff-administered medications, particularly for complex regimens like parenterals, are more impractical given the high costs and disruptions associated with regular home visits [55].

Nevertheless, barriers for self-administration exist due to patients' physical or cognitive disabilities. Strategies including risk assessments, strong HaH team support, and streamlined processes should be implemented to assess patient suitability and boost patient confidence [54]. When self-administration is not feasible, clear processes must be established for staff intervention. Effective communication and accurate record-keeping are essential for staff-administration to prevent duplication or omission [32]. Additionally, contractual agreements should be incorporated into care planning to define responsibilities and accountabilities [28].

## Case-in-point: Singapore

In Singapore, where HaH remains relatively new [56–59], our recommendations align with national objectives to reduce healthcare costs and optimise resource utilisation [60]. However, planners should consider local standards to enhance patient safety and health outcomes.

Home environment suitability remains critical for HaH inclusion in Singapore. Many homes face spatial limitations challenging proper storage of medical supplies and safe patient care [61]. Further patient stratification based on home suitability is recommended to address these challenges. While Housing Development Board initiatives to integrate medical services into estates show promise, short-term solutions should consider these spatial constraints [62].

Strengthening relationships with community providers like Homage and telemedicine networks for post-discharge care support clinical handovers and external collaborations [63–65]. Initiatives like the National Electronic Health Record and 'One Patient, One Health Record' enhance patient data sharing and healthcare communication, supporting an integrated care network [66]. However, partnerships with GPs may face challenges due to opportunity costs of leaving clinics unattended and the travel time involved [67]. Leveraging productivity-enhancing technologies is crucial to optimise interface time with patients and mitigate opportunity costs.

A proficient multidisciplinary care team is essential for holistic HaH care. Local initiatives fostering cross-disciplinary collaboration in public hospitals can extend to HaH services [68]. Training healthcare professionals in clinical skills and psychological support is vital, as exemplified by Singapore's GeriCare programme, which equips nursing homes to manage acute and subacute conditions [69].

By integrating these recommendations and addressing local challenges, Singapore can enhance the efficacy and sustainability of its HaH implementation, ultimately enhancing care for its population.

## Guideline quality

The quality of guideline methodology varied, with only four recommended for use in their current form. The remaining ten were recommended with modifications. Overall, guidelines scored lowest in 'Rigour of Development' and 'Editorial Independence'.

'Rigour of Development' assesses the methodology used for locating, synthesising, developing, and updating recommendations. Poor scores in this domain indicate potential weaknesses in the evidence base and methodological transparency, suggesting that adherence to these guidelines may not ensure optimal outcomes for HaH patients. Most guidelines

leveraged on expert opinion during their development, and this lack of methodological rigour or transparency may have contributed to the notably low scores here.

Regarding 'Editorial Independence', guidelines should disclose funding sources, their influence on guideline development, and potential conflicts of interest. Most guidelines failed to provide adequate information on these aspects, which can undermine their credibility and raise concerns about possible biases in recommendation formulation. This lack of editorial oversight may reflect the fact that many HaH guidelines are developed by national authorities or professional bodies with limited external scrutiny or independent validation.

Despite low scores in these domains, all fourteen guidelines were still deemed appropriate for information extraction due to their practical value, alignment with long-standing HaH programme experiences, and input from key stakeholders or opinion leaders in their development. Many HaH guidelines are designed primarily for practical implementation that focus on outlining objectives, target audiences, and recommendations. This emphasis on usability likely resulted in lower scores in less directly applicable domains. Nevertheless, the absence of standardized, high-quality development processes highlights the need for more rigorous HaH guidelines.W hile AGREE II offers a structured framework for evaluating guideline methodology, it does not inherently assess the quality and comprehensiveness of guideline content. Guidelines developed with rigorous processes may not offer universally acceptable recommendations, while those with less formalized methodologies may still offer valuable, practice-oriented guidance.

## Limitations

This study has limitations. Firstly, guidelines not addressing acute care in geriatric, internal medicine, and general medicine specialties were excluded, omitting valuable recommendations from HaH programmes in other specialities. As a result, the findings may be more applicable to broad HaH implementation but less relevant to specialty-specific models that could be crucial in settings with more unique, complex, or patient-specific needs. While the study provides a comprehensive foundation for general HaH frameworks, its potentially narrower study impact on the wider medical practice must be acknowledged. Future research should explore specialty-driven HaH models to ensure a more inclusive approach that accommodates diverse clinical pathways and patient populations.

Secondly, due to translation restrictions, relevant publications in other languages were excluded. This limited the geographical coverage and comprehensiveness of the recommendations. The HaH models analysed were primarily from countries with well-established programmes, which may not fully represent the implementation challenges faced in lower-resource settings or regions with different regulatory frameworks. Additionally, the exclusion of countries like Taiwan and France, despite their established HaH programs, resulted in the loss of potentially valuable insights. As a result, the findings may be more applicable to high-income healthcare systems or English-speaking countries, where healthcare structures may share greater similarities. Future research should foster international collaboration to develop broader, more inclusive guidelines that account for diverse healthcare landscapes.

Lastly, guideline updates may lag, potentially overlooking new primary literature evidence that could enhance recommendations at the time of this review. Emerging models or post-pandemic innovations in HaH may not be fully reflected in these findings. Future studies should integrate ongoing developments in HaH care, ensuring that recommendations remain adaptive to new healthcare challenges and advancements.

## Conclusion

In conclusion, this literature review analysed fourteen guidelines on HaH implementation, aiming to consolidate information and address conflicting guidance. HaH planners can use these recommendations, adapting them to meet programme-specific requirements, with the goal of ensuring high-quality patient care amidst evolving healthcare challenges and patient needs.

The long-term sustainability of HaH programmes hinges on deliberate policy shifts across financing, regulation, digital health, and workforce, alongside strategic and equitable resource allocation. To ensure its success, HaH must transition from being viewed as a pilot or alternative model to becoming a mainstream, preferred component of national healthcare infrastructure. By embedding HaH within broader health systems, policymakers can enhance its scalability, ensuring that it delivers effective, patient-centered care in the long run.

## Supporting information

**S1 Table.  Search syntax.**
(DOCX)

**S2 Appendix.  Component data table.**
(XLSX)

**S3 Table.  PRISMA 2020 checklist.**
(DOCX)

**S4 Table.  Analysed database literature.**
(XLSX)

## Author contributions

**Conceptualization:** Valerie Ng, Yi Feng Lai.

**Data curation:** Clive Goh, Jun Jie Soh.

**Formal analysis:** Clive Goh, Jun Jie Soh, Valerie Ng, Yi Feng Lai.

**Investigation:** Clive Goh, Jun Jie Soh.

**Methodology:** Clive Goh, Jun Jie Soh.

**Project administration:** Yi Feng Lai.

**Supervision:** Valerie Ng.

**Validation:** Clive Goh, Jun Jie Soh, Valerie Ng, Yi Feng Lai.

**Visualization:** Clive Goh, Valerie Ng, Yi Feng Lai.

**Writing – original draft:** Clive Goh, Jun Jie Soh.

**Writing – review & editing:** Clive Goh, Jun Jie Soh, Valerie Ng, Yi Feng Lai.

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
