## [Decision Letter · Decision Letter 0]

PONE-D-24-48516Hospital-at-Home care in Singapore: a review of overseas protocols and guidelines to support implementation and policy redesign (systematic review)PLOS ONE

Dear Dr. Goh,

Thank you for submitting your manuscript to PLOS ONE. After careful consideration, we feel that it has merit but does not fully meet PLOS ONE’s publication criteria as it currently stands. Therefore, we invite you to submit a revised version of the manuscript that addresses the points raised during the review process.

We look forward to receiving your revised manuscript.

Kind regards,

Shrikanth Gadad

Academic Editor

PLOS ONE

**Journal Requirements:**

2. As required by our policy on Data Availability, please ensure your manuscript or supplementary information includes the following: 

Reviewers' comments:

Reviewer's Responses to Questions

**Comments to the Author**

1. Is the manuscript technically sound, and do the data support the conclusions?

Reviewer #1: Yes

Reviewer #2: Yes

Reviewer #3: Yes

2. Has the statistical analysis been performed appropriately and rigorously? 

Reviewer #1: Yes

Reviewer #2: Yes

Reviewer #3: Yes

3. Have the authors made all data underlying the findings in their manuscript fully available?

Reviewer #1: Yes

Reviewer #2: Yes

Reviewer #3: Yes

4. Is the manuscript presented in an intelligible fashion and written in standard English?

Reviewer #1: Yes

Reviewer #2: Yes

Reviewer #3: Yes

5. Review Comments to the Author

**Reviewer #1:** The manuscript is entitled "Hospital-at-Home Care in Singapore: A Review of Overseas Protocols and Guidelines to Support Implementation and Policy Redesign." The following suggestions are designed to further strengthen the clarity, rigor, and applicability of your work, which addresses a very important aspect of healthcare delivery.

1. The objective should be concise and directly aligned with the study's core intention. Adding a brief mention of the AGREE II scoring range in the results summary provides helpful context regarding the quality of the evaluated guidelines. Additionally, explaining potential implications for policy redesign in Singapore can strengthen your conclusion and make your findings more applicable to the local healthcare setting.

2. It is highly recommended that the study protocol be registered on PROSPERO to minimize bias.

3. Briefly explain why the five regions (Australia, Canada, Spain, the UK, and the USA) were selected for your review. This clarification will help contextualize their relevance to Singapore's healthcare system.

4. Please clarify whether the data extraction process was piloted or tested before full implementation. Piloting can improve consistency and minimize extraction errors. Additionally, detailing the measures taken to reduce selection bias during screening will enhance the reliability of your findings.

5. A more detailed comparison of high-scoring and low-scoring guidelines would offer valuable insights into what characterizes higher-quality guidelines. Visual aids such as graphs or heat maps illustrating the AGREE II scores could improve accessibility and understanding of the data.

6. Expanding on how your findings can be integrated into Singapore’s existing healthcare framework will enhance the study’s relevance. Providing clear, practical recommendations for policy implementation would be particularly beneficial.

In the discussion, consider elaborating on your study's limitations and their potential impact on the generalizability of the findings. This transparency helps contextualize the results and acknowledges any constraints that may affect their applicability.

**Reviewer #2: **The manuscript provides a very useful overview of HaH guidelines from different countries. The use of the AGREE II tool to assess guideline quality is laudable. However, several limitations hinder the comprehensiveness of the scope and methodological rigor of the study. To further enhance the quality and impact of the manuscript, the following recommendations are made below:

1. A clearly pre-defined protocol with registration details is missing. It will enhance transparency while strictly adhering to methodological standards.

2. To broaden the findings' global applicability, consider including non-English guidelines and those from various healthcare specialties.

3. Further research is needed on how guideline gaps in implementing HaH affect patient outcomes.

4. Discuss and propose new models or strategies likely to improve the implementation challenges and enhance patient care.

5. Critically assess the guidelines' quality, especially those with a low score regarding "Rigor of Development" and "Editorial Independence."

6. Describe the process for resolving disagreements in AGREE II ratings between reviewers.

Interviews or surveys to capture the perspectives of patients, caregivers, and healthcare providers.

7. Discuss the long-term policy implications and resource allocation strategies necessary for sustaining HaH programs.

8. To better understand HaH's potential, the broader applications outside acute care should be discussed in the discussion section of the manuscript

**Reviewer #3: **In the systematic overview “Hospital-at-Home care in Singapore: a review of overseas protocols and guidelines to support implementation and policy redesign”, the author was looking to create a set of guidelines that healthcare providers worldwide could use to implement HaH. The author believes this kind of overview is needed due to a lack of effective implementation guidelines. By doing a literature review across several databases and internet, they were able to put together guidelines which, according to AGREE II, were the most appropriate for implementation of HaH. The results showed a wide range of guideline quality, however, they were able to come up with a distinct set of recommendations which could be used in Singapore and worldwide.

One minor concern could be the small number of publications actually used. While the original number of publications evaluated was quite large, the final set was quite small. Could additional databases be used to add more literature? Or why a smaller data set was used?

Will another overview be done for specialty care? It is mentioned that this overview is covering the general population, and not considering specialty care, what was the reasoning behind that?

Another concern in the quality assessment is that the AGREEII tool recommends at least 2, preferably 4 appraisers, is there a particular reason why only 2 were used?

(ii) Number of Appraisers We recommend that each guideline be assessed by at least 2 appraisers, and preferably 4, as this will increase the reliability of the assessment.)

6. PLOS authors have the option to publish the peer review history of their article (what does this mean?). If published, this will include your full peer review and any attached files.

Reviewer #1: **Yes: **Bharathi Gadad

Reviewer #2: **Yes: **Nikhilesh Anand

Reviewer #3: No

---

## [Author Response · Author response to Decision Letter 1]

1 Apr 2025

1st April 2025

Dr. Shrikanth Gadad

Academic Editor

PLOS ONE

Dear Dr. Shrikanth,

Original Research Study Manuscript Re-Submission

Hospital-at-Home care in Singapore: a review of overseas protocols and guidelines to support implementation and policy redesign (systematic review)

We would like to re-submit the above-titled manuscript revision for consideration of publication in your esteemed journal as an original clinical research study article.

This manuscript was previously submitted, reviewed and recommended for resubmission (Manuscript PONE-D-24-48516). We would like to thank the reviewers for their valuable comments. We have since gone through the suggestions in detail and our responses are as follow:

The comments of the reviewer(s) are included here:

Reviewer #1: The manuscript is entitled "Hospital-at-Home Care in Singapore: A Review of Overseas Protocols and Guidelines to Support Implementation and Policy Redesign." The following suggestions are designed to further strengthen the clarity, rigor, and applicability of your work, which addresses a very important aspect of healthcare delivery.

1. The objective should be concise and directly aligned with the study's core intention. Adding a brief mention of the AGREE II scoring range in the results summary provides helpful context regarding the quality of the evaluated guidelines. Additionally, explaining potential implications for policy redesign in Singapore can strengthen your conclusion and make your findings more applicable to the local healthcare setting.

Response:

Objective:

- We provided more information on AGREE II in the revised manuscript to provide further context and aid in the interpretation of AGREE II results in the abstract.

Potential implications for policy redesign in Singapore:

- We agree with this suggestion. More information was added in the revised abstract to further explain potential implications and to strengthen the conclusion. More information on the potential implications for policy redesign in Singapore can be seen in the section under “Case in point: Singapore”

2. It is highly recommended that the study protocol be registered on PROSPERO to minimize bias.

Response: While the study protocol was not registered prior to the conduct of this review, the PRISMA standards were adhered to strictly and the methods were clearly stated in the manuscript.

3. Briefly explain why the five regions (Australia, Canada, Spain, the UK, and the USA) were selected for your review. This clarification will help contextualize their relevance to Singapore's healthcare system.

Response: We agree with this point. More information has been added to the revised manuscript, under the section on ‘Data Extraction”.

4. Please clarify whether the data extraction process was piloted or tested before full implementation. Piloting can improve consistency and minimize extraction errors. Additionally, detailing the measures taken to reduce selection bias during screening will enhance the reliability of your findings.

Response: The data extraction process was neither piloted nor tested before full implementation.

- Two independent reviewers conducted the data extraction. The extracted datasets were compared and any discrepancies/differences were resolved through consensus. This involved an online or physical meeting with both reviewers to discuss and raise up any points of contention within the research articles selected.

5. A more detailed comparison of high-scoring and low-scoring guidelines would offer valuable insights into what characterizes higher-quality guidelines. Visual aids such as graphs or heat maps illustrating the AGREE II scores could improve accessibility and understanding of the data.

Response: Table 3 illustrates the breakdown of AGREE II scores for all fourteen guidelines. Articles that were considered ‘high quality’ are highlighted in blue, as seen in the table. The individual domain scores can also be gleaned from the table, and would show that ‘high quality’ guidelines had higher scores in a larger majority of domains compared to those deemed ‘low quality’.

As AGREE II has no set standard for what is deemed as ‘high quality’, a 50% cut off was established based on a literature review of cut off scores used by research articles that relied on AGREE II for quality assessment. Information on the papers can be retrieved from citations 25 and 26, and further explained under the section on “Quality assessment”.

Finally, the designation of ‘low quality’ vs ‘high quality’ was eventually deemed irrelevant for the research paper given the nature of the information contained within the selected guidelines. AGREE II domains such as conflict of interest or funding were not relevant for the selected guidelines as 13 of the 14 were developed by government/state/health authorities, where such information may not be as crucial compared to proper research papers. Hence, all fourteen guidelines were eventually deemed suitable for extraction (regardless of their quality) as they had provided sufficient information that were useful for practical information.

6. Expanding on how your findings can be integrated into Singapore’s existing healthcare framework will enhance the study’s relevance. Providing clear, practical recommendations for policy implementation would be particularly beneficial.

In the discussion, consider elaborating on your study's limitations and their potential impact on the generalizability of the findings. This transparency helps contextualize the results and acknowledges any constraints that may affect their applicability.

Response: “…how your findings can be integrated into Singapore’s existing healthcare framework … clear, practical recommendations for policy implementation…”

- The section “Case in point: Singapore” illustrates how the findings can be integrated into Singapore’s existing healthcare framework. No practical recommendations for policy implementation were provided for the local context as the focus of this paper is to compile, compare, and provide overall best practices for HaH implementation from a global context. Thereafter, these insights can then be gleaned and adapted to Singapore’s healthcare system, as applicable.

“… study’s limitations and their potential impact on the generalizability…”

- While a ‘limitations’ section was included in the earlier draft, it is agreed that the potential impacts were not fully elucidated. More information has been included in the ‘limitations’ section to highlight potential impacts on the generalisability of the findings.

-----

Reviewer #2: The manuscript provides a very useful overview of HaH guidelines from different countries. The use of the AGREE II tool to assess guideline quality is laudable. However, several limitations hinder the comprehensiveness of the scope and methodological rigor of the study. To further enhance the quality and impact of the manuscript, the following recommendations are made below:

1. A clearly pre-defined protocol with registration details is missing. It will enhance transparency while strictly adhering to methodological standards.

Response: While the study protocol was not registered prior to the conduct of this review, the PRISMA standards were adhered to strictly and the methods were clearly stated in the manuscript.

2. To broaden the findings' global applicability, consider including non-English guidelines and those from various healthcare specialties.

Response: Non-English guidelines were excluded due to a lack of translation resources, as explained in the ‘data extraction’ section.

Other healthcare specialities were excluded to focus on more broad-based/general recommendations for HaH programmes. The objective of the paper aimed at covering HaH programmes for the general population, before considering specialty care. This was explained in the ‘search strategies section, and further evaluated in the ‘limitations’ section.

- New information was added in the ‘Limitations’ section to address any potential implications on the applicability of our findings.

3. Further research is needed on how guideline gaps in implementing HaH affect patient outcomes.

Response: The research aims to compile/streamline recommendations and reconcile any differences.

- Further research on the effect of guideline gaps on patient outcomes is not the main aim of the research. Nevertheless, some potential impacts on patient outcomes were addressed in relevant sections under ‘Discussion’.

4. Discuss and propose new models or strategies likely to improve the implementation challenges and enhance patient care.

Response: Recommendations to resolve any implementation challenges or conflicting recommendations were similarly also highlighted in the ‘Discussion’ and ‘Case in point: Singapore” sections.

5. Critically assess the guidelines' quality, especially those with a low score regarding "Rigor of Development" and "Editorial Independence."

Response: Further elaboration was provided under the section on “Guideline quality”.

6. Describe the process for resolving disagreements in AGREE II ratings between reviewers.

Interviews or surveys to capture the perspectives of patients, caregivers, and healthcare providers.

Response: The process for AGREE II ratings first involved independently assigning scores to each domain item in AGREE II, before coming together to review each item individually. Any differences or discrepancies in scores exceeding two points were resolved through consensus discussions.

7. Discuss the long-term policy implications and resource allocation strategies necessary for sustaining HaH programs.

Response: “The long-term sustainability of HaH programmes hinges on deliberate policy shifts across financing, regulation, digital health, and workforce, alongside strategic and equitable resource allocation. To ensure its success, HaH must transition from being viewed as a pilot or alternative model to becoming a mainstream, preferred component of national healthcare infrastructure.”

- The above views have also been included in the discussions/conclusion sections of the paper.

8. To better understand HaH's potential, the broader applications outside acute care should be discussed in the discussion section of the manuscript

Response: Non-acute care patients were excluded as it falls outside the scope of care provided by the HaH model defined in this study:

- “Hospital-at-Home (HaH) is a care model delivering acute, hospital-level care to patients in their own homes, typically for a specified duration, as a substitute for acute inpatient admission”.

---

Reviewer #3: In the systematic overview “Hospital-at-Home care in Singapore: a review of overseas protocols and guidelines to support implementation and policy redesign”, the author was looking to create a set of guidelines that healthcare providers worldwide could use to implement HaH. The author believes this kind of overview is needed due to a lack of effective implementation guidelines. By doing a literature review across several databases and internet, they were able to put together guidelines which, according to AGREE II, were the most appropriate for implementation of HaH. The results showed a wide range of guideline quality, however, they were able to come up with a distinct set of recommendations which could be used in Singapore and worldwide.

Q: One minor concern could be the small number of publications actually used. While the original number of publications evaluated was quite large, the final set was quite small. Could additional databases be used to add more literature? Or why a smaller data set was used?

Response: The two databases: Scopus and Web of Science, were initially chosen because they are prominent and multidisciplinary databases covering sciences, medicine, and social sciences, etc. This helped to expand the scope of our research. However, the papers retrieved from both databases mainly published study results on e.g., the effectiveness, potential issues, cost effectiveness, stakeholder perceptions, etc of HaH. These papers lacked recommendations.

After independently reviewing the literature from the databases and resolving any discrepancies via consensus discussion, all but one guidelines was removed according to our inclusion/exclusion criteria. The inclusion/exclusion criteria specified that papers must provide “recommendations of specific guidelines, standard-operating-procedures, and/or policies for implementation of HaH”.

Due to this difference, we expected a low output from our database search. Rather than increase the number of databases used, we chose instead to supplement our searches with the Advanced Internet Search. We also noticed that other guideline-based systematic reviewers included Advanced Internet Searches (in this case, search engines like Google) to supplement or increase the amount of literature retrieved. These search engines often provided more detailed guidelines programme frameworks for HaH as they were typically created by government or institutional sources, meant for distribution between states or across health networks or hospitals.

Q: Will another overview be done for specialty care? It is mentioned that this overview is covering the general population, and not considering specialty care, what was the reasoning behind that?

Response: Our research question aimed at compiling guidelines from internal, general, and geriatric medicine specialties. The rationale was to focus on more general medical cases/needs rather than zooming in on specific disease area/organ/demographic. We intend to optimise the current model of HaH implementation for the general population before branching out into specialised patient groups.

- More information can be found in the ‘search strategies’ and ‘limitations’ sections.

- Further information was also added in the ‘limitations’ section to address the potentially narrower study impact on the wider medical practice.

Q: Another concern in the quality assessment is that the AGREEII tool recommends at least 2, preferably 4 appraisers, is there a particular reason why only 2 were used? (ii) Number of Appraisers We recommend that each guideline be assessed by at least 2 appraisers, and preferably 4, as this will increase the reliability of the assessment.)

Response: Due to resource constraints, two appraisers were used for the AGREE II assessment instead of the recommended four. However, both appraisers were involved in the majority of the research/writing, with the senior authors serving as research mentors and tie breakers.

To enhance robustness of the results, consensus discussions were conducted to resolve any discrepancies.

---

## [Decision Letter · Decision Letter 1]

Hospital-at-Home care in Singapore: a review of overseas protocols and guidelines to support implementation and policy redesign (systematic review)

PONE-D-24-48516R1

Dear Dr. Goh,

We’re pleased to inform you that your manuscript has been judged scientifically suitable for publication and will be formally accepted for publication once it meets all outstanding technical requirements.

Kind regards,

Shrikanth Gadad

Academic Editor

PLOS ONE

Additional Editor Comments (optional):

Reviewers' comments:

Reviewer's Responses to Questions

**Comments to the Author**

1. If the authors have adequately addressed your comments raised in a previous round of review and you feel that this manuscript is now acceptable for publication, you may indicate that here to bypass the “Comments to the Author” section, enter your conflict of interest statement in the “Confidential to Editor” section, and submit your "Accept" recommendation.

Reviewer #2: All comments have been addressed

Reviewer #3: All comments have been addressed

2. Is the manuscript technically sound, and do the data support the conclusions?

Reviewer #2: Yes

Reviewer #3: Yes

3. Has the statistical analysis been performed appropriately and rigorously? 

Reviewer #2: Yes

Reviewer #3: Yes

4. Have the authors made all data underlying the findings in their manuscript fully available?

Reviewer #2: Yes

Reviewer #3: Yes

5. Is the manuscript presented in an intelligible fashion and written in standard English?

Reviewer #2: Yes

Reviewer #3: Yes

6. Review Comments to the Author

Reviewer #2: (No Response)

Reviewer #3: (No Response)

7. PLOS authors have the option to publish the peer review history of their article (what does this mean?). If published, this will include your full peer review and any attached files.

Reviewer #2: **Yes: **Nikhilesh Anand

Reviewer #3: No

---

## [Editor Report · Acceptance letter]

PONE-D-24-48516R1

PLOS ONE

Dear Dr. Goh,

I'm pleased to inform you that your manuscript has been deemed suitable for publication in PLOS ONE. Congratulations! Your manuscript is now being handed over to our production team.

Kind regards,

on behalf of

Dr. Shrikanth Gadad

Academic Editor

PLOS ONE